# Antimicrobial Resistance of *Erysipelothrix rhusiopathiae* Strains Isolated from Geese to Antimicrobials Widely Used in Veterinary Medicine

**DOI:** 10.3390/antibiotics12081339

**Published:** 2023-08-19

**Authors:** Kamila Bobrek, Andrzej Gaweł

**Affiliations:** Department of Epizootiology and Clinic of Birds and Exotic Animals, Faculty of Veterinary Medicine, Wrocław University of Environmental and Life Sciences, 50-366 Wrocław, Poland; kamila.bobrek@upwr.edu.pl

**Keywords:** *Erysipelothrix rhusiopathiae*, geese, antimicrobial resistance

## Abstract

The aim of this study was to determine the antibiotic resistance of *E. rhusiopathiae* when isolated from clinical outbreaks of erysipelas in geese to antimicrobials commonly used in poultry production. All isolates were susceptible to amoxicillin alone or with clavulanic acid, with MIC values ranging from 0.016 to 0.125 μg/mL. Ninety-six percent of isolates were fully sensitive to penicillin G (MIC 0.125–0.5 μg/mL). All isolates were fully or moderately sensitive to erythromycin (MIC 0.125–0.5 μg/mL). Most *E. rhusiopathiae* isolates proved resistant to fluoroquinolones (76.6% of isolates were resistant to enrofloxacin, with MIC values ranging from 0.064 to 32 μg/mL, and 68% were resistant to norfloxacin, with MIC values ranging from 0.094 to 96 μg/mL), and tetracyclines (61.7% of isolates were resistant to doxycycline, with MIC values ranging from 0.25 to 64 μg/mL, and 63.8% were resistant to tetracycline, with MIC values ranging from 0.38 to 256 μg/mL). Point mutations in the gyrA gene (responsible for fluoroquinolone resistance) and the presence of the tetM gene (responsible for tetracycline resistance) were noted in most of the resistant isolates. Multidrug resistance, defined as resistance to at least one substance in three or more antimicrobial classes, was not observed.

## 1. Introduction

*Erysipelothrix rhusiopathiae* is a small Gram-positive rod that causes erysipelas in many animal species, particularly in swine and turkeys [1]. Erysipelas in pigs can occur as an acute septicemic or chronic disease, with the development of arthritic lesions and endocarditis [2]. In pigs, the disease is controlled by vaccination and, in the case of clinical outbreaks, antibiotic therapy. Commercial vaccines contain inactivated bacteria (most commonly serotype 2) and are believed to provide high protection against serotype 1 and 2 strains, which cause the majority of clinical cases [3]. Erysipelas in other species is less frequent. *E. rhusiopathiae* causes polyarthritis in lambs and calves, and it is also an important pathogen in the field of public health. In humans, infections can present as a localized erysipeloid, in the diffuse cutaneous form, or as bacteremia progressing to infective endocarditis [4,5,6]. For many years, erysipelas outbreaks of economic significance were not often reported in poultry species other than turkeys [7]. However, the epidemiological situation has changed, and reports describing erysipelas outbreaks in chickens, hens, and geese have been described [8,9,10]. Our research shows that erysipelas could be a problem in geese flocks due to the rearing system in which birds can have contact with contaminated soil. *E. rhusiopathiae*, which causes septicemia and high mortality in geese flocks, and during necropsy, petechiae on the heart muscle, pancreas, and liver can be observed [11]. Prevention is limited to antibiotic treatment due to the lack of registered vaccines for geese. Currently, the data on *E. rhusiopathiae* isolated from poultry are scarce. If the birds are not properly treated, the bacteria may develop resistance to the antibiotics used and then spread, not only leading to treatment failures in birds but also meaning that these resistant strains could be transferred to humans.

The aim of this study was to determine the antibiotic resistance of *E. rhusiopathiae* strains isolated from clinical outbreaks of erysipelas in geese flocks for the most commonly used antimicrobials in poultry production.

## 2. Results

### 2.1. Isolation and Identification of Erysipelothrix Strains

All 47 isolates of Gram-positive rods cultured from geese septicemia and collected between 2008 and 2018 were identified by PCR with genus- and species-specific primers as *Erysipelothrix rhusiopathiae* [Table 1]. In all erysipelas cases, the monoculture of *E. rhusiopathiae* was confirmed without the contamination of any other pathogen. No *E. tonsillarum*, *Erysipelothrix* sp. strain 1, or *Erysipelothrix* sp. strain 2 were confirmed.

### 2.2. Serotyping of E. rhusiopathiae Isolates

The most prevalent serotype among the *E. rhusiopathiae* geese isolates was serotype 1b, which comprised 26 isolates (55.3%). Thirteen isolates (27.7%) belonged to serotype 2, and eight isolates (17%) belonged to serotype 5. No isolates were identified as belonging to the 1a serotype [Table 2].

### 2.3. Antimicrobial Susceptibility Test

All *E. rhusiopathiae* isolates were susceptible to amoxicillin (AC) and amoxicillin with clavulanic acid (XL), with MIC values ranging from 0.016 to 0.125 μg/mL. Additionally, most of the isolates (96%) were fully susceptible to penicillin G (PG), but two isolates showed intermediate susceptibility with an MIC value of 0.19μg/mL. The MIC_50_ value for penicillin was 0.032 μg/mL, and the MIC_90_ value was 0.094 μg/mL. The majority of isolates (85%) were fully susceptible to erythromycin (EM), and seven isolates showed intermediate MIC values ranging from 0.125 to 0.5 μg/mL with an MIC_50_ of 0.19 μg/mL and an MIC_90_ of 0.5 μg/mL.

Most isolates proved to be resistant to fluoroquinolones and tetracyclines. Resistance to enrofloxacin (EF) was observed in 76.6% of isolates, with MIC values ranging from 0.064 to 32 μg/mL, and resistance to norfloxacin (NX) was identified in 68% of isolates, with MIC values ranging from 0.094 to 96 μg/mL. The MIC_50_ values were 32 μg/mL for enrofloxacin and 24 μg/mL for norfloxacin, and the MIC_90_ values for enrofloxacin and norfloxacin were 32 μg/mL and 48 μg/mL, respectively.

In total, 61.7% of isolates were resistant to doxycycline (DC), with MIC values ranging from 0.25 to 64 μg/mL, and 63.8% were resistant to tetracycline (TC), with MIC values ranging from 0.38 to 256 μg/mL. For doxycycline, the MIC_50_ value was 24 μg/mL, and the MIC_90_ value was 48 μg/mL, while for tetracycline, the MIC_50_ value was 24 μg/mL, and the MIC_90_ value was 64 μg/mL.

Multidrug resistance, defined as resistance to at least one substance in three or more antimicrobial classes [15], was not observed. Among the 47 isolates, 9 (19.1%) were susceptible to all tested antimicrobials. Among the 47 *E. rhusiopathiae* isolates, 5 phenotypic resistance patterns were observed. Most of the isolates (26 isolates; 55.3%) were resistant to doxycycline, tetracycline, enrofloxacin, and norfloxacin (DC–TC–EF–NX). Two isolates were resistant to doxycycline, enrofloxacin, and norfloxacin (DC–EF–NX), two to fluoroquinolones (EF–NX), one to tetracyclines (DC–TC), and three to tetracycline (TC). Four isolates showed intermediate susceptibility to tetracyclines. The multiple antimicrobial resistance (MAR) indexes ranged between 0 and 0.5.

### 2.4. Detection of Resistance Genes

Several of the obtained resistance profiles showed phenotypic resistance to tetracyclines and fluoroquinolones, and the majority of the genotypic resistance profiles were compatible with phenotypic resistance. The tetM gene was detected in 38 isolates (80.9%). In four isolates, the presence of the tetM gene did not cause phenotypic resistance to doxycycline and tetracycline, and two isolates were resistant to tetracycline while remaining susceptible to doxycycline. Two isolates were resistant to tetracycline and doxycycline without the presence of the TetM gene. The gyrA gene was amplified in all the samples, and the product was sequenced to determine whether a mutation at position 257 was present.

### 2.5. Sequence and Analysis of the Selected PCR Products

Among all sequenced gyrA gene products, 19 showed T, 17 showed C, and 11 showed A at position 257. The presence of C at this position indicated fluoroquinolone susceptibility, while the presence of A or T indicated resistance [16]. The presence of A or T at position 257 was compatible with the observed phenotypic resistance. Most of the isolates with C (88.2%) were susceptible to enrofloxacin and norfloxacin. One isolate showed resistance to both substances, while one was susceptible only to enrofloxacin and one only to norfloxacin.

The sequencing of products with genus-, species-, and serotype-specific primers confirmed the accuracy of PCR reactions. The tetM and gyrA PCR products were consistent with the database. No unique sequences were identified in any of the PCR reactions.

## 3. Discussion

Erysipelas in birds is generally characterized as an acute, fulminating infection. It is a septicemic disease; however, its chronic form occasionally occurs after acute outbreaks. The disease usually occurs suddenly, with only a few birds found dead initially, followed by increasing levels of mortality on subsequent days. Mortality usually ranges from <1% to 50% [8,17,18] and is higher in flocks that keep in contact with the external environment, such as on domestic or ecological farms, in aviaries, or with access to pastures [19]. Data on the characteristics of the *E. rhusiopathiae* strains causing erysipelas in geese are missing; thus, our research provides new information on this topic. We showed that there is a limited variety of serotypes in *E. rhusiopathiae* isolates from geese, and only three serotypes, 1b, 2, and 5, were noted in this investigation, which is consistent with previous data on poultry isolates [20,21,22]. We found no phenotypic multidrug resistance among *E. rhusiopathiae* isolates from geese. The drugs of first choice in cases of animal erysipelas include penicillin; however, the latest data on drug susceptibility testing of *E. rhusiopathiae* strains from poultry showed the presence of isolates resistant to this group of antimicrobials. Hess et al. [20] identified multidrug-resistant isolates resistant to penicillin G and ampicillin in hen and turkey *E. rhusiopathiae* collected between 2003 and 2021. Our research did not confirm the presence of penicillin-resistant isolates of *E. rhusiopathiae* in geese, and there was no multidrug resistance among the isolates. All isolates were susceptible to amoxicillin, which is one of the most often used antimicrobials in poultry production and the drug of first choice in erysipelas outbreak treatments in birds [19]. The susceptibility of geese isolates to penicillins was consistent with that of swine isolates from various regions of the world [23,24,25,26,27,28]. A similar status of geese and swine isolates susceptible to erythromycin (the antimicrobial often used in erysipeloid treatment) was noted. No resistance to erythromycin was found in geese and pig isolates in Poland [24]; however, studies have reported that the drug susceptibility of *Erysipelothrix* strains to this antimicrobial is variable. The presence of intermediate-sensitivity isolates noted in our own research was consistent with previous reports from Japan [25,29]. Contrasting results were recorded in China, where increasing numbers of macrolide-resistant *Erysipelothrix* strains isolated from pigs have been confirmed. Erythromycin-resistant isolates in poultry were also detected [11,20].

In contrast to previous research on *E. rhusiopathiae*‘s susceptibility to fluoroquinolones [27,28], our research confirmed the latest reports on the increasing resistance to this antimicrobial group [20,24]. The high prevalence of enrofloxacin (76.6%) and norfloxacin (68%) resistant isolates were reported in this study, and the MIC values were distributed in two ranges, similar to the Chuma report [23], qualifying *E. rhusiopathiae* as susceptible or resistant strains. The reason for the high prevalence of enrofloxacin-resistant strains could be due to the frequent use of this drug in geese production and mutations in the gyrase gene (gyrA): the enzyme that alters DNA supercoiling [30]. The gyrA gene sequences obtained from geese isolates in most resistant isolates showed point mutations at position 257 (C→T or C→A), changing the threonine amino acid to isoleucine or lysine, which has been demonstrated to confer resistance [16,24]. However, we observed phenotypic resistance in several isolates without point mutations in the gyrA gene; therefore, we suspect that mutations in the gyrA gene are only one mechanism of drug resistance in *E. rhusiopathiae.* In Gram-positive bacteria, fluoroquinolones have two targets of action: DNA gyrase and topoisomerase IV, both of which are necessary for DNA replication. Different fluoroquinolones have different levels of inhibitory activity against the two enzymes, and, in several cases, topoisomerase IV appears to be more sensitive [30,31]. Our research confirmed that the dominant cause of fluoroquinolone resistance is a mutation in the gyrA gene; however, it showed that other causes of drug resistance could not be excluded.

*Erysipelothrix* isolates from geese showed varied resistance to tetracyclines, which is consistent with previous reports [23,25,27]. This resistance in bacteria can be caused by one of the following mechanisms: inactivation, efflux by proton antiporters, the protection of ribosomes, or genetic resistance [25]. Previous reports have shown that, in *E. rhusiopathiae*, tetracycline resistance is mainly caused by genetic resistance—such as tetM gene’s presence—that has the ability to block the binding of and/or displace tetracycline from the ribosomal 30S subunit [16,24,25,32]. Our research showed the wide dissemination of the tetM gene in *E. rhusiopathiae* from geese in over 80% of isolates. Not all resistant isolates were positive for the tetM gene, which suggests the presence of other effective resistance mechanisms against tetracyclines.

In conclusion, our research showed that all *E. rhusiopathiae* in this study were penicillin-susceptible isolates; however, the increasing resistance of *E. rhusiopathiae* to antimicrobials widely used in veterinary medicine requires further monitoring. The high prevalence of isolates resistant to fluoroquinolone confirmed the presence of mutations in the gyrA gene, and resistance to tetracyclines was confirmed by the detection of the tetM gene.

## 4. Materials and Methods

### 4.1. Isolation and Identification of Erysipelothrix Strains

*Erysipelothrix* strains were isolated post-mortem from geese with lesions typical for septicemia, where erysipelas was suspected. Necropsies and bacteriological investigations were performed at the Department of Epizootiology with the Clinic of Birds and Exotic Animals, Wroclaw Environmental and Life Sciences, in cooperation with poultry veterinarians. Swabs were taken from damaged organs, such as the liver, heart, and lungs, and were cultured on blood agar (Graso, Owidz, Poland) aerobically at 37 °C for 48 h. Small grayish-translucent colonies with alpha hemolysis zones were taken for Gram staining to confirm the presence of slender Gram-positive rods.

To confirm that the analyzed strains belonged to the *E. rhusiopathiae* species, genomic DNA was extracted using the Genomic Mini Kit (A&A Biotechnology, Gdynia, Poland) and according to the manufacturer’s instructions for PCR amplification with genus-specific MO primers and species-specific ER primers [Table 1]. The PCR reactions were performed in a 25 µL reaction mixture containing 5 ng of template DNA, 10 pmol of each primer, 12.5 µL of the PCR Mix Plus (1.25 U) (A&A Biotechnology, Gdynia, Poland), and ultrapure water. The reaction with the MO primers was carried out as follows: initial denaturation for 5 min at 94 °C, followed by 30 cycles consisting of denaturation for 1 min at 94 °C, annealing for 1 min at 54 °C, and extension for 1 min at 72 °C, with a final elongation for 7 min at 72 °C. The multiplex PCR reactions with ER primers were carried out as follows: initial denaturation for 5 min at 94 °C, followed by 35 cycles consisting of denaturation for 1 min at 94 °C, annealing for 1 min at 58 °C, and extension for 1 min at 72 °C, with final elongation for 10 min at 72 °C using an iCycler (Biorad Laboratories Inc., Hercules, CA, USA). The PCR products were subjected to electrophoresis in 1.5% agarose gel stained with SYBR Green (Sigma Aldrich, Poznan, Poland). The presence of products ~400 bp in both reactions confirmed that the isolates belonged to *Erysipelothrix rhusiopathiae.* As positive controls, *E. rhusiopathiae* IW445 and *E. tonsillarum* IW779 from the National Veterinary Research Institute’s (Puławy, Poland) collection were used. As negative controls, *E. coli* ATCC 25922 and *S. aureus* ATCC 29213 were used. The 47 confirmed *E. rhusiopathiae* strains collected between 2008 and 2018 were preserved and stored at −80 °C in 2 mL tubes with 40% glycerol/10 mL for future tests.

### 4.2. Serotyping of E. rhusiopathiae Strains

Serotyping was performed based on multiplex PCR reactions performed in a 25 µL reaction mixture containing 25 ng of template DNA, 10 pmol of each primer [Table 1], 12.5 µL of PCR Mix Plus (1.25 U) (A&A Biotechnology, Gdynia, Poland), and ultrapure water. The reaction was carried out as follows: initial denaturation for 2 min at 95 °C, 35 cycles consisting of denaturation for 30 s at 95 °C, annealing for 30 s at 60 °C and extension for 1 min at 72 °C, with final elongation for 10 min at 72 °C. The PCR products were subjected to electrophoresis in 1.5% agarose gel stained with SYBR Green (Sigma Aldrich, Poznan, Poland).

### 4.3. Antimicrobial Susceptibility Test

To determine the minimum inhibitory concentrations (MICs) for selected chemotherapeutics, E-test (bioMerieux, Warszawa, France) strips with a defined gradient of chemotherapeutic concentrations (15 double sequential dilutions) were used.

For this study, we selected E-tests containing substances used in poultry medicine with potentially high activity against *E. rhusiopathiae*, including penicillin G, amoxicillin, amoxicillin with clavulanic acid, enrofloxacin, norfloxacin, tetracycline, doxycycline, and erythromycin.

Due to the observed poor growth on Mueller–Hinton agar supplemented with 5% defibrinated sheep blood, the strains were cultured on tryptose–soy agar supplemented with 5% defibrinated sheep blood [23]. The E-tests were used according to the methodology provided by the manufacturer. Incubation was carried out for 24 h at a temperature of 37 °C, after which the MIC value was read at the point of intersection for the resulting elliptical growth inhibition zone with the edge of the strip. The quality controls of the antimicrobial substances were carried out using *E. coli* ATCC 25922 and *S. aureus* ATCC 29213.

The results were adapted and evaluated according to the Clinical Laboratory Standards Institute’s standards [33,34]. The minimum inhibitory concentration (MIC) breakpoints for penicillin G were ≤0.125 susceptible, and for amoxicillin and amoxicillin with clavulanic acid, they were ≤0.25 susceptible. The MIC breakpoints for erythromycin were ≤0.25 susceptible and ≥1 resistant. The breakpoints for fluoroquinolones were ≤0.5 susceptible and ≥2 resistant. The breakpoints for tetracyclines were ≤4 susceptible and ≥16 resistant. After determining the MIC values for each of the antibiotics and for all isolated strains, population analyses were carried out to determine the MIC_50_ and MIC_90_ values [35]. The multiple antibiotic resistance (MAR) index was calculated as the quotient between the number of antimicrobials to which isolates were resistant to the number of tested antimicrobials [36].

Multidrug resistance was defined as resistance to at least one substance in three or more antimicrobial classes.

### 4.4. Detection of Resistance Genes

PCR reactions based on gene-specific primers for tetracyclines (TetM 5′-GTGGACAAAGGTACAACGAG-3′; 5′-CGGTAAAGTTCGTCACACAC-3′) and fluoroquinolones (gyrA 5′-TCGTCTCCTATGCCATGTCG-3′; 5′-AGTAAAAGTGCCCCTGTTGGA-3′) were performed in a 50 µL reaction mixture using 50 ng of the DNA template, 20 pmol of each primer, and 25 µL of PCR Mix Plus (1.25 U) and ultrapure water. The thermal profile for both genes was as follows: initial denaturation for 5 min at 95 °C, followed by 30 amplification cycles of a denaturing step at 95 °C for 50 s, annealing at 54 °C for 40 s, and extension at 72 °C for 1 min, with a final extension cycle at 72 °C for 10 min [37]. The PCR products were subjected to electrophoresis in 1.5% agarose gel stained with SYBR Green (Sigma Aldrich, Poznan, Poland).

### 4.5. Sequence and Analysis of the Selected PCR Products

To confirm the accuracy of the PCR results, the representative three products of the reaction with genus-, species-, and serotype-specific primers were selected for sequencing. Among the resistance gene products, five randomly chosen tetM and all gyrA gene products were selected. The PCR products were isolated from agarose gels using a Gel Out Concentrator Kit (A&A Biotechnology, Gdynia, Poland) and were subsequently sent to Macrogen (Amsterdam, The Netherlands) for Sanger sequencing with both forward and reverse PCR primers. The sequences were analyzed using Mega X software ver. 10.2.5 and were compared to sequences from the National Center for Biotechnology Information (NCBI) GenBank database.

## Figures and Tables

**Table 1 antibiotics-12-01339-t001:** Primers for PCR reactions identifying the genus, species, and serotype of *Erysipelothrix*.

	Primer Name	Sequence	Targeting	Product Size [bp]	References
Primers for genus and species identification	MO101-MO102	5′-AGATGCCTAGAAACTGTA-3′ 5′-CTGTATCCGCCATAACTA-3′	*Erysipelothrix* spp.	407	[12]
	ER1F-ER1R	5′-GTTCATCTCTCTAATGCACTAC-3′ 5′-TGTTGGACTACTAATCGTTTCG-3′	*E. rhusiopathiae*	339	[13]
	ER2F-ER2R	5′-ATGTAATATGATCTGGTGATTTG-3′ 5′-AGGACTGCTGATTGTCTCATG-3′	*E. tonsillarum*	384	[13]
	ER3F-ER3R	5′-TGGAGGACCGAACCGACTG-3′ 5′-AATTTTGGGACCTTAACTGGC-3′	*Erysipelothrix* sp. strain 1	289	[13]
	ER4F-ER4R	5′-TAAAGCACTAAGATCTGGTGG-3′ 5′-TCGGACTACTAATTGTCTCAG-3′	*Erysipelothrix* sp. strain 2	387	[13]
Primers for serotype identification	1aF-1aR	5′-CTCCTAACGCTTTAGCACGC-3′ 5′-TGATCCTTTGCCACTAATGC-3′	*E. rhusiopathiae* serotype 1a	356	[14]
	1bF-1bR	5′-CGAAAGCATCCCTTAATCATTGC-3′ 5′-TGCGTGTAAAACCTGATCGTGTAAATC-3′	*E. rhusiopathiae* serotype 1b	1357	[14]
	2F-2R	5′−CCACGTCTTCCCACACTACAAAAAAGTAAATTC-3′ 5′- TCATCCTAATGCATATCATTATGTGGATATGAA-3	*E. rhusiopathiae* serotype 2	541	[14]
	5F-5R	5′-GCACGTTTCCAAATATTGTATCGAGTCT-3′ 5′-GAAATAATGCCGATAGATGGAGCACC-3′	*E. rhusiopathiae* serotype 5	194	[14]

**Table 2 antibiotics-12-01339-t002:** Characteristics of *Erysipelothrix rhusiopathiae* isolates.

		Phenotypic Antimicrobial Resistance									Genetic Antimicrobial Resistance	
Isolate No	Serotype	Penicillin			Tetracyclines		Fluoroquinolones		Macrolides	MAR Index	Tetracyclines	Fluoroquinolones
		AC	XL	PG	DC	TC	EF	NX	EM		tetM Gene Presence	gyrA Point Mutations
1	1b	0.016	0.016	0.032	**16**	12	**32**	**64**	0.38	0.375	+	T
2	2	0.016	0.016	0.047	0.25	**16**	0.125	0.094	0.125	0.125	−	C
3	1b	0.032	0.032	0.032	0.5	0.38	0.125	0.38	0.19	0	+	C
4	2	0.064	0.064	0.047	0.94	1.5	0.25	0.125	0.125	0	+	C
5	2	0.032	0.032	0.006	**32**	**32**	**32**	**48**	0.19	0.5	+	T
6	5	0.032	0.032	0.012	**34**	**96**	**12**	**32**	0.125	0.5	+	A
7	1b	0.047	0.047	0.047	1	1	**32**	0.19	0.125	0.125	+	C
8	1b	0.023	0.023	0.023	0.75	0.5	0.094	0.125	0.19	0	−	C
9	1b	0.032	0.032	0.016	**24**	**32**	**32**	**48**	0.19	0.5	+	T
10	5	0.047	0.047	0.003	**32**	**24**	**32**	**32**	0.25	0.5	+	A
11	1b	0.016	0.016	0.19	12	8	**12**	**48**	**0.5**	0.25	+	T
12	2	0.094	0.094	0.032	0.38	3	0.5	0.125	0.19	0	−	C
13	1b	0.032	0.032	0.025	**32**	**32**	**32**	**48**	0.23	0.5	−	T
14	1b	0.032	0.032	0.032	**32**	**48**	**32**	**48**	0.25	0.5	+	T
15	1b	0.023	0.023	0.047	**48**	16	**32**	**24**	0.19	0.375	+	A
16	2	0.064	0.064	0.064	0.5	0.75	0.064	0.125	0.064	0	−	C
17	5	0.032	0.032	0.032	**48**	**48**	**32**	**48**	0.125	0.5	+	T
18	2	0.023	0.023	0.047	**48**	**256**	**32**	**48**	0.19	0.5	+	T
19	1b	0.047	0.047	0.012	**32**	**16**	**32**	**24**	0.5	0.5	+	C
20	2	0.047	0.047	0.012	0.75	0.38	0.064	0.38	0.25	0	+	C
21	2	0.023	0.023	0.047	**48**	**24**	**32**	**32**	0.5	0.5	+	T
22	5	0.064	0.064	0.025	**24**	**24**	**32**	**12**	0.38	0.5	+	A
23	1b	0.047	0.047	0.094	**32**	**32**	**32**	**64**	0.5	0.5	+	A
24	1b	0.047	0.047	0.032	**32**	**32**	0.25	0.19	0.125	0.25	+	C
25	1b	0.064	0.064	0.094	**24**	**24**	**32**	**24**	0.19	0.5	+	T
26	2	0.064	0.064	0.012	**16**	**24**	**32**	**96**	0.19	0.5	+	A
27	1b	0.032	0.032	0.016	**24**	**24**	**32**	**32**	0.19	0.5	+	T
28	5	0.094	0.094	0.047	**24**	**32**	**32**	**48**	0.25	0.5	+	T
29	1b	0.064	0.064	0.012	**24**	**32**	**32**	**48**	0.25	0.5	+	T
30	1b	0.125	0.125	0.047	0.75	**24**	0.125	0.125	0.25	0.125	+	C
31	5	0.016	0.016	0.016	0.75	**256**	0.094	**48**	0.125	0.25	+	A
32	1b	0.016	0.016	0.006	**32**	**32**	**32**	**24**	0.38	00.5	+	A
33	1b	0.064	0.064	0.016	**24**	**24**	**32**	**32**	0.19	0.5	−	T
34	1b	0.064	0.064	0.025	**64**	**64**	**32**	**48**	0.5	0.5	+	T
35	1b	0.094	0.094	0.032	**16**	0.38	0.125	0.125	0.5	0.125	+	C
36	1b	0.125	0.125	0.047	0.75	0.38	0.125	0.125	0.25	0	−	C
37	1b	0.032	0.032	0.006	**32**	**24**	**32**	**48**	0.25	0.5	+	A
38	2	0.032	0.032	0.032	1	0.38	0.125	0.125	0.19	0	−	C
39	2	0.016	0.016	0.032	**16**	12	**32**	**64**	0.25	0.375	+	A
40	5	0.023	0.023	0.047	**32**	**16**	**32**	**24**	0.19	0.5	+	T
41	1b	0.016	0.016	0.19	12	8	**12**	**32**	0.125	0.25	+	T
42	1b	0.023	0.023	0.016	**48**	**64**	**32**	**32**	0.5	0.5	+	T
43	2	0.094	0.094	0.032	0.38	0.75	0.5	0.125	0.19	0	−	C
44	5	0.032	0.032	0.032	12	0.38	0.125	0.38	0.125	0	+	C
45	1b	0.064	0.064	0.025	**24**	**32**	**32**	**12**	0.125	0.5	+	T
46	2	0.032	0.032	0.006	12	12	0.064	0.125	025	0	+	C
47	1b	0.032	0.032	0.016	**24**	**32**	**32**	**32**	0.19	0.5	+	A

AC—amoxicillin; XL—amoxicillin with clavulanic acid; PG—penicillin G; DC—doxycycline; TC—tetracycline; EF—enrofloxacin; NX—norfloxacin; EM—erythromycin; MAR—index multiple antimicrobial resistance index. The minimal inhibitory concentration (µg/mL) for intermediate strains are highlighted grey and resistant strains are grey and bold.

## Data Availability

Not applicable.

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
