# Peer review of "Antimicrobial Resistance of Erysipelothrix rhusiopathiae Strains Isolated from Geese to Antimicrobials Widely Used in Veterinary Medicine"

_antibiotics, 2023, doi:10.3390/antibiotics12081339_

Round 1
Reviewer 1 Report
The introduction of this manuscript is relevant and sufficient information about the previous study findings is presented for readers to follow the present study rationale and procedures. The methods are appropriate. The results are clear, explained with appropriate statistics. The discussion is appropriate; the authors make a systematic contribution to the research literature in this area of investigation. However, many critical issues and editing errors emerged along with the manuscript that has to cover in order to improve its readability and scientific sound. Point-by-point comments are listed below:
In line 103, reference should be 10, 13.
The current language is not enough for publication in this journal, so the language needs further improvement.
The background, significance and purposes of this study need to be discussed in more detail.
The manuscript has some grammatical mistakes; the author should revise it thoroughly, especially in introduction and discussion section.
The discussion is not in-depth enough and needs to be compared with some previous studies.
Adding the limitations of the manuscript in the “Discussion or Conclusion” will make the research more interesting.
Quality of English Language is good
Author Response
Dear Reviewer,
Thank you for considering our manuscript for publication and providing us with your comments. We added the analysis of serotyping and genetic resistance to tetracyclines and quinolones, because among the isolates those to groups showed diversity in phenotypic resistance. We also discussed more detailed the need of this research, and we hope that it is in-depth enough now.
Thank you for the comment about the language, you were not only person who wrote that it needs improvement. We decide to send our manuscript to professional scientific language editing in MDPI (the certificate included). The refrences were changed.
We hope that the corrected text will find your approval.
AG and KB

Reviewer 2 Report
I consider the submitted manuscript to be an interesting short communication, touching on important topics in the field of veterinary medicine and microbiology. The epidemiology of Erysipelothrix rhusiopathiae among geese and the antibiotic resistance of strains is a very important report and an innovative approach. The most frequently studied is the pathogenicity and occurrence of Erysipelothrix rhusiopathiae among pigs, however, erysipelas is of great importance in goose breeding, which is addressed in this work. Monitoring and control of antibiotic resistance among zoonotic bacteria is particularly important because they can be a source of the spread of genes conditioning resistance not only to antibiotics but also to chemotherapeutic agents, disinfectants and other biocides.
I rate the publication very well, it is an excellent introduction to the development of research. There are minor linguistic and punctuation errors in the text, which, however, do not affect the value of the work. All comments were made by the reviewer on the text of the manuscript, the file is attached.

Author Response
Dear Reviewer,
Thank you for considering our manuscript for publication and providing us with your comments. We decide to added the analysis of serotyping and genetic resistance to tetracyclines and quinolones, because among the isolates those to groups showed diversity in phenotypic resistance.
Thank you for the comment about the minor linguistic and punctuation errors. We removed the repetitions, nad rearranged suggested sentences. We decide to send our manuscript to professional scientific language editing in MDPI to improve the language (the certificate included).
We hope that the corrected text will find your approval.
AG and KB

Reviewer 3 Report
This article determine the antibiotic resistance of E. rhusiopathiae strains isolated from clinical outbreaks of erysipelas in geese to the antimicrobials commonly used in poultry production. i have following quries that need to be adressed before final publication.
Q: 01 If all the isolates were susceptible to amoxicillin then why amoxicillin in combination with clavulanic acid was used?
Q: 02 Amoxicillin and penicillin G belongs to same microbial class B- lactam antibiotic? Why both were selected from same class?
Q: 03 What could be the reason that different strains provided different MIC value against same antibiotic e.g; erythromycin?
Q: 04 Which drug was least susceptible against isolate?
Q: 05 Fluoroquinolones is a class of antibiotic with different agents. How you will justify not to use all Fluoroquinolones in poultry?
Q: 06 Does all swabs from septicemia contained only erysipelothrix strain no any other pathogen was observed?
Q: 07 Reference strain used in study were obtained from where? Purchased or obtained from which laboratory mention name?
Q: 08 Doxycycline is a tetracycline antibiotic why checked differently for susceptibility?
Q: 09 If resistance among these classes has not reported then why these class of antibiotics selected in study?
Q: 10 Specie specific primer should be written clearly that were used in PCR ?
Some minor changes required
Author Response
Dear Reviewer,
Thank you for considering our manuscript for publication and providing us with your comments. We added the analysis of serotyping and genetic resistance to tetracyclines and quinolones, because among the isolates those to groups showed diversity in phenotypic resistance.
The penicillins group is widely used in poultry production in Poland, because of wide antimicrobial spectrum and also good penetration to tissues. The most commonly used antimicrobial is amoxicillin. We took 3 antimicrobials to see, if there is any difference between the often used amoxicillin and penicillin G which is used rare. Some veterinarians observed that more useful in erysipelas in geese treatment is amoxicillin with clavulanic acid, so we want to check this observation in vitro (Q1,Q2). We observed that the susceptibility for different antimicrobials differed beetween the isolates, what could be natural feature when the source of infection is not the same (Q3). Among the antimicrobials which should be effective against E.rhusiopathiae, two groups - tetracyclines and fluoroquinolones were the less effective, because of genetic resistance of the isolates (Q4). The fluoroquinolones should be used wisely, only with antibiograms and in correct dose to be sure that the treatment will be effective (Q5). We added the information that the pure culture of E.rhusiopatiae was obtained from erysipelas cases (Q6), and the information about the reference strains and sequencing (Q7). Doxycycline is more often use in poultry, so we planned to check if there is any difference between the resistance. As you can see, the MIC values are in many isolates different, but generally, if isolate was resistant for doxycycline, it was also resistant for tetracycline. We decide not to describe the antimicrobials which are known as unuseful in erysipelas treatment (colistin or aminoglycosides) - we checked that all srtains were resistant to those antimicrobials, but it was described in literature as a natural resistance, so we decide not to show it. The varied susceptibility to penicillins, noted among E.rhusiopatie isolated from poultry was th reason why we decide to investigate it more widely (Q9). The primers were described in the new table (Tab1) (Q10)
We also decide to send our manuscript to professional scientific language editing in MDPI (the certificate included) to improve the language.
We hope that the corrected text will find your approval.
AG and KB

Reviewer 4 Report
In this study, the authors described the antimicrobial profile of Erysipelothrix rhusiopathiae isolates responsible for Erysipelas in geese.
General comments:
- the manuscript should be classified as a Short communication;
- the authors did not perform the clonal characterization of the isolates through molecular methods - as such, the term "strains" should be corrected to "isolates" throughout the manuscript, as it is not possible to conclude if the isolates under study correspond (or not) to different strains;
- extensive editing of english language is required, beggining with the title, which should be changed to "Antimicrobial resistance of Erysipelothrix rhusiopathiae isolates from geese to antimicrobials widely used in veterinary medicine". Other examples: lines 39-41 (how can the lack of data be a source of antibiotic-resistant isolates to humans?); lines 88-89; lines 102-103; lines 109-110; line 149;
- a review of the formatting of all the text is required for the correction of some minor issues (example, m;l in line 15 or the missing space between E. and rhusiopathiae in line 40; fluoroquionolones in line 115; ºC in material and methods);
- it must be clear in the conclusions that the results correspond to the group of isolates under study, and not to all E. rhusiopathiae linked to geese infections.
Other comments:
- Title of Table 1 must be corrected, as it should be representative of the information presented in the table. To facilitate results interpretation, resistant and intermediate values must be highlighted in different colours; MDR isolates should also be highlighted;
- Calculate and discuss the isolates antimicrobial resistance (MAR) index (see as reference Matos, A.; Cunha, E.; Baptista, L.; Tavares, L.; Oliveira, M.
ESBL-Positive Enterobacteriaceae from Dogs of Santiago and Boa Vista
Islands, Cape Verde: A Public Health Concern. Antibiotics 2023, 12, 447.
https://doi.org/10.3390/antibiotics12030447;
- Line 79 - The fact that 3 isolates are tetracycline resistan does not correspond to a resistance pattern. Correct this sentence;
- Lines 88-89 - Are these isolates MDR or just resistant to penicillin G and ampicillin?;
- Line 124 - Change "changed organs" to "organs with lesions";
- Line 124-125 - The best samples for E. rhusiopathiae isolation are the liver, spleen, kidney, heart and synovial tissue. Why were the lungs sampled?
- Lines 125-127 - In E. rhusiopathiae, alpha hemolysis only appears after 48 hours of incubation, yet the authors state that only a 24 hours incubation was performed.
- Lines 130-142 - What was the control used for PCR? If none, PCR products must be sequenced for results confirmation. Check the protocol for volumes (ml?). Which were the amplification conditions in the first and last step of the PCR? Which were the conditions applied in electrophoresis? How was the PCR product revealed? A picture of a electrophoresis gel must be included in the manuscript.
- Lines 153-154 - Which guidelines were used to select the medium, CLSI or EUCAST? Change the reference by the proper guideline.
- Lines 155-156 - E-tests are bought as ready-to-use stripes, as such are not made by the authors. Correct.
- Lines 160-161 - Specify the adaptations made to the guidelines methodology.
- Lines 161-165 - Very confusing, transform that information in a Table.
- Lines 167-169 - This sentence must be supported by a reference (Magiorakos, A. P., Srinivasan, A., Carey, R. B., Carmeli, Y., Falagas, M. E., Giske, C. G., Harbarth, S., Hindler, J. F., Kahlmeter, G., Olsson-Liljequist, B., Paterson, D. L., Rice, L. B., Stelling, J., Struelens, M. J., Vatopoulos, A., Weber, J. T., & Monnet, D. L. (2012). Multidrug-resistant, extensively drug-resistant and pandrug-resistant bacteria: an international expert proposal for interim standard definitions for acquired resistance. Clinical microbiology and infection : the official publication of the European Society of Clinical Microbiology and Infectious Diseases, 18(3), 268–281. https://doi.org/10.1111/j.1469-0691.2011.03570.x).
Extensive editing of english language is required.
Author Response
Dear Reviewer,
Thank you for considering our manuscript for publication and providing us with your comments. We added the analysis of serotyping and genetic resistance to tetracyclines and quinolones, because among the isolates those to groups showed diversity in phenotypic resistance. Thank you for the comment about the language, you were not only person who wrote that it needs improvement. We decide to send our manuscript to professional scientific language editing in MDPI (the certificate included). We agree that word "strain" should be changed in our manuscript to "isolate" - it was done. We also changed in conlusions the sentence about the E.rhusiopathiae- we hope that its clear now that isolates in this study had those features.
The table 1 was modified according your suggestions, MAR index was added. Poultry organs in which E.rhusiopathiae was detect were liver, heart, splean, but also lungs because the disease is in acute form,as sepsis. From kidneys we did not isolate the bacterium, maybe because of the uric acid presence (kidneys are not typical organ for bacteria isolations in poultry).
We corrected the sentences which you suggested, and we hope the text will find your approval.
AG and KB

Round 2
Reviewer 1 Report
It could be accepted in the current version.
It could be accepted in the current version.
Reviewer 3 Report
I am satisfied with revided version and would like to accept this article for publication